# Assessment of Physical Performance in Children After Surgery for Congenital Diaphragmatic Hernia

**DOI:** 10.3390/jcm13237249

**Published:** 2024-11-28

**Authors:** Anna Pałka, Bogumiła Strumiłło, Anna Piaseczna-Piotrowska

**Affiliations:** Clinic of Pediatric Surgery and Urology, Polish Mother’s Health Center Institute, ul. Rzgowska 281/289, 93-310 Łódź, Polandannapiaseczna@yahoo.com (A.P.-P.)

**Keywords:** congenital diaphragmatic hernia (CDH), motor skills, long-term outcomes

## Abstract

**Objective**: This study aims to evaluate long-term physical and motor performance in children aged 3–6 years following congenital diaphragmatic hernia (CDH) surgery, in comparison with healthy peers. While existing research emphasizes prognostic factors such as the lung to heart ratio (LHR) and liver position, few studies address physical outcomes in early childhood post-surgery. **Methods**: A total of 31 children who underwent neonatal CDH surgery (study group) and 41 age-matched healthy children (reference group) were assessed. The Wrocław Test and the Ozierecki Metric Scale (modified by Barański) were used to evaluate strength, endurance, motor coordination, and agility. **Results**: Findings indicated that children in the CDH group had significantly lower scores in physical performance, particularly in endurance and motor coordination, compared to their healthy peers. Strength differences were present but less pronounced. **Conclusions**: Children post-CDH surgery exhibit slightly delayed physical and motor development, suggesting a potential need for targeted physical activity programs in early childhood to support improved outcomes.

## 1. Introduction

Congenital diaphragmatic hernia (CDH) is a rare but serious developmental defect, occurring at a frequency of 1 to 5 cases per 10,000 births [1,2]. This condition is characterized by the displacement of abdominal organs into the thoracic cavity through a diaphragmatic defect, leading to impaired lung development and other severe complications [2]. Thanks to the use of modern prenatal and postnatal therapies, as well as a better understanding of the pathophysiology of pulmonary hypertension and lung hypoplasia, the survival rate of patients, even those with severe forms of the defect, has increased and currently ranges from 55% to 98% [1,3]. This increase in survival has led to a higher incidence of complications, both those resulting from treatment and those related to the nature of the defect. These complications may include the following [4,5,6]:Respiratory system (due to lung hypoplasia and prolonged mechanical ventilation: bronchopulmonary dysplasia, recurrent respiratory infections, and asthma);Digestive system (gastroesophageal reflux, adhesive bowel obstruction, hiatal hernia);Skeletal system (chest deformities);Psycho-somatic developmental delays;Low body weight;Hearing impairment;Recurrent diaphragmatic hernia.

In addition to these medical complications, CDH can impact physical growth and motor skill development in affected children. Recent studies have suggested that children with CDH may experience delays in achieving physical milestones and may have lower endurance and motor coordination compared to their healthy peers [7,8,9]. Most reports in the literature focus on early postoperative outcomes, the search for risk factors, and their impact on the survival of children with CDH. Given the range of long-term complications, there is a need for further research to evaluate the physical and motor development of these patients as they grow, especially in the context of quality of life and the potential for targeted rehabilitation therapies.

The aim of this study is to assess the physical performance, endurance, and motor coordination of children after congenital diaphragmatic hernia surgery and compare their results with those of a group of healthy peers. The study aims to provide data that may contribute to the development of recommendations for long-term care of patients after CDH surgery.

## 2. Materials and Methods

This study included a total of 72 children aged 3 to 6 years—study group (*n* = 31) and control group (*n* = 41). Between 2016 and 2019, 83 patients with congenital diaphragmatic hernia (CDH) were born at the Polish Mother’s Health Center Institute. Of these, 48 patients (57.83%) died, with 20 (41.67%) passing away immediately after birth, before they could undergo surgical intervention. The remaining 28 (58.33%) patients died due to cardiorespiratory failure, pulmonary hypertension, and other congenital anomalies. Thirty-five patients (42.17%) survived and were discharged from the hospital. The study group consisted of children who underwent neonatal surgery for congenital diaphragmatic hernia at the Polish Mother’s Health Center Institute in Łódź. An additional inclusion criterion for this group was the absence of accompanying genetic, neurological, or musculoskeletal disorders that could influence the physical performance assessment [5,6,7]. The control group (*n* = 41) included healthy children attending Municipal Kindergarten No. 4 “Misia Uszatka” in Pabianice. Written consent was obtained from parents and the kindergarten director. The study was approved by the Research Ethics Committee of the Polish Mother’s Health Center Institute in Łódź (approval no. 4/2019).

Physical examination and anthropometric measurements were conducted for both groups, including body weight, height, head circumference, and chest circumference. Data were plotted on percentile charts to assess growth patterns.

To assess physical performance, the following validated assessment tools tests were conducted:

Wrocław Test—a standard and widely accepted test for assessing children’s physical performance which covers four main components. It has been shown in previous studies to exhibit high reproducibility and reliability in the pediatric population [10,11,12,13,14]:Strength test—throwing a 1 kg medicine ball from above the head. Each child performed 2 practice throws and 3 official throws. The longest distance was recorded and converted into points based on the test result table. The test was conducted under the supervision of qualified specialists who ensured proper guidance and support during the exercise. Each child’s response to the test was carefully monitored, and any signs of discomfort or excessive strain would have led to the immediate termination of the test. This approach prioritized the safety and well-being of participants, aligning with recommended practices for strength assessment in young children.Power test—standing long jump. Each participant performed 1 practice jump and 3 official jumps. The longest jump was recorded and converted into points according to the result table.Speed test—a 20 m sprint from a standing start. Each child performed the task twice. The faster result was recorded and converted into points based on the test result table.Agility test—shuttle run 4 × 5 m (envelope) with block transfer, modified from the International Physical Fitness Test. The task was performed twice. The better result was recorded and points were awarded according to the tables.

Each test was conducted twice, and the better result was taken for analysis. The results were entered into score tables, considering gender and age. The total points were summed, and the results were classified according to the Wrocław Test scoring system:

Single trial score:Up to 39 pts.—low, insufficient level;40–49 pts.—sufficient level;50–59 pts.—good level;60 pts. and above—high, very good level.

Overall test score:Up to 159 pts.—low, insufficient physical performance;160–199 pts.—sufficient physical performance;200–239 pts.—good physical performance;240 pts. and above—high, very good physical performance.

### 2.1. Ozierecki Metric Scale, Modified by Barański

The scale, validated in previous studies, was designed to assess six fundamental motor behaviors in each age group, providing high reliability for motor skills assessment in the pediatric population. In Groups I and II, three-year-old children were present. Due to their active participation and willingness to engage in the study, exercises from the set for four-year-olds were conducted with them, although their results were not included in the statistics. Given their young age, these children might not have met the required thresholds, which could result in unreliable outcomes [15,16], (Table 1):I.Static coordination (balance)—assessed through static balance exercises;II.Dynamic coordination of hands—assessed through manipulative exercises, throwing a ball at a target, and catching a thrown ball;III.Dynamic coordination of the whole body—assessed through exercises such as jumps, leaps, and skips;IV.Movement speed—primarily exercises involving the hands, but also those checking the ability to perform a series of consecutive tasks;V.Simultaneous movements—simultaneous movements of both hands or parallel performance of multiple tasks;VI.Precision of movements—assessing the ability to perform precise movements without unnecessary additional movements (synkinesis).

For correct task execution, the child received a “+”; for partially correct execution, a “±”; and for incorrect or incomplete execution, a “−”. Each “+” added 2 months to the child’s motor age, and each “±” added 1 month. The test began by determining the child’s chronological age to the year, month, and day, followed by the age-appropriate test. If the child successfully completed the tasks, tests for older age groups were conducted. The test ended when the child could not complete any task from the next test set. If the child did not complete all tasks for their age group, exercises for younger children were conducted to determine the set where the child obtained only “+”.

To ensure the reproducibility of assessments, each test was conducted twice, and the better result was used for analysis to minimize variability. All test results were normalized according to age-specific standards, allowing for precise interpretation in the context of participants’ ages. Separate scoring tables were applied for each age category (3, 4, 5, and 6 years), which enabled an accurate assessment of physical and motor performance based on age-appropriate norms. The measurement tools used, such as the Wrocław Test and the Ozierecki Metric Scale modified by Barański, are recognized and standardized methods for assessing physical and motor performance in children across different ages. These tools account for developmental differences, allowing for reliable evaluation within the framework of physical and motor development. All physical performance assessments were conducted indoors under controlled conditions to ensure consistency and minimize the influence of environmental factors on test outcomes.

**Table 1 jcm-13-07249-t001:** Ozierecki Tests according to the Barański modification.

No. of the Test (Exercise Group)	Age (Years)	
	4	5
I	Balance stance in a lunge, right foot toes touching the left foot heel, eyes closed. Hold for 15 s.	Balance stance on high tiptoes, eyes open. Hold for 10 s.
II	Touch the tip of the nose with the index finger of both the right and left hand in a direct motion. Eyes closed.	Using only one hand, crumple a 5 × 5 cm napkin. Completion time: right hand 15 s, left hand 20 s.
III	Jump in place using both feet for 5 s. The task is considered completed if the take-off was from both feet and the jumps were performed on the toes.	Hop on one leg along a 5 m line. Do not deviate more than 5 cm from the line. Perform on both the right and left leg.
IV	Place 20 coins in a box. Task time: 20 s.	Wind a 2 m thread into a ball. Task time: right hand 12 s, left hand 18 s.
V	Simultaneously draw circles outward with the index fingers of both hands. Arms extended forward. Task time: 10 s	Using both hands simultaneously, place 10 matches into a box. Task time: 20 s.
VI	Shake the examiner’s hand with both the left and right hand. Attention is paid to synkineses.	Grit your teeth. Attention is paid to synkineses.
**Test No.** **(Exercise Group)**	**Age (years)**	
	**6**	**7**
I	Balance stance on one leg, the other leg bent at a 90° angle, thigh pointing downward. Hold for 10 s. Perform on both the right and left leg.	Balance stance leaning forward while simultaneously standing on tiptoes. Hold for 10 s.
II	Throw a tennis ball from a distance of 1.5 m at a 25 × 25 cm target; boys: 3 throws, girls: 4 throws. Task completion: Right hand: at least 2 accurate throws Left hand: at least 1 accurate throw	Draw the path through mazes with the right and left hand. Task time: Right hand: 1 min Left hand: 2 min
III	Jump with both feet, without a run-up, over a rope stretched at a height of 20 cm.	Measure a distance of 2 m with your feet, along a straight line, without deviating from the line.
IV	Draw vertical lines between the lines of legal paper (accuracy + 3 mm). Task time: 10 s. Task completion:	Taking one card at a time, divide a deck of 36 cards into 4 piles. Task duration: Right hand: 30 s
	Right hand: 20 lines Left hand: 12 lines	Left hand: 40 s
V	While walking around the room, wind a thread around your fingers. Task time: 15 s. Do not change the walking pace.	Tap a steady rhythm with one hand, while the other hand draws circles in the air with the index finger. Do not change the rhythm.
VI	Strike the table with a percussion hammer. Attention is paid to synkineses.	Raise the eyebrows. Attention is paid to synkineses.
**Test No.** **(Exercise Group)**	**Age (years)**	
	**8**	**9**
I	Balance stance in a squat, arms out to the sides, eyes closed. Task time: 10 s.	Balance stance on the left and right leg, eyes closed. Task time: 10 s.
II	Touch the tips of each finger on the same hand sequentially with the tip of the index finger. Task time: 5 s.	Boys: hit a 25 × 25 cm target with a tennis ball from a distance of 2.5 m. Right hand: 5 throws, at least 3 accurate; Left hand: 5 throws, at least 2 accurate; Girls: cut circles from paper. Task time: right hand 1 min, left hand 1 min 30 s
III	Using one leg, as in hopscotch, push a box between two lines over a distance of 5 m.	Boys: jump without a run-up, using both feet, over a height of 40 cm.
	The distance between the lines is 50 cm.	Girls: jump up and clap at least 3 times simultaneously above the head.
IV	The participant runs 5 m to a table, takes 4 matches out of a closed box, arranges them into a square, folds a sheet of paper in half, and returns to their starting position. Task time: 15 s.	Turn the pages of a book with the right and left hand. Task time: 15 s. Right hand: 30 pages; Left hand: 18 pages.
V	While seated, simulate marching. When stepping with the right foot, tap the table with the index finger of the right hand. Do not change the rhythm.	While seated, simulate marching at any pace. Simultaneously with stepping the right foot, tap the table with both index fingers. Do not change the rhythm.
VI	Frown your forehead. Attention is paid to synkineses.	While sitting on a chair, lift your legs to a height of 20–25 cm. Bend one leg, then the other, at the ankle joint as quickly as possible.

### 2.2. Statistical Analysis

Measurable features were described using measures of central tendency—arithmetic mean and median—and measures of dispersion—standard deviation, 95% confidence interval, and the minimum and maximum values of the analyzed variables.

A skewness and kurtosis test was initially conducted to assess the normality of the distribution of the measurable variable, and Levene’s test was used to estimate the homogeneity of variances. A multifactorial analysis of variance (ANOVA) without repetitions was performed for variables with a normal distribution. For variables with a distribution deviating from normality, generalized linear models with robust standard errors were applied. All these multivariate models were controlled for gender, age, body weight, and height of the participants.

For categorical variables, the Chi-square independence test, Fisher’s exact test (in cases of small cell counts in contingency tables), and log-linear analysis were used. A significance level of *p* < 0.05 was considered statistically significant. All statistical calculations and significance tests were performed using Statistica™ software, version 14 (TIBCO Software Inc., Palo Alto, CA, USA).

To account for the natural differences in physical development among children aged 3 to 6 years, the results of physical and anthropometric tests were analyzed within the context of age groups. Additionally, data were normalized according to age-specific standards for each test, allowing for precise interpretation of results relative to participants’ ages.

## 3. Results

### Demographic and Anthropometric Characteristics of the Studied Children

The mean age of all children participating in the study was M = 4.35 years. The majority of children included in the study were girls—39 (52.00%). However, when considering the group affiliation separately, boys constituted the majority in the study group—17 (54.84%)—while girls were the majority in the reference group—25 (56.82%).

The mean head circumference for both groups was 51.40 cm (SD = 3.16), with no statistically significant difference between Group I and Group II (*p* = 0.644). Children in Group I had a smaller mean chest circumference (52.94 cm, SD = 4.30) than those in Group II (54.93 cm, SD = 3.68), which was statistically significant (*p* = 0.036). Mean body weight and height for the entire sample were 18.62 kg (SD = 3.80) and 110.97 cm (SD = 9.39), respectively, with no significant differences between groups.

On percentile charts, the median head circumference percentiles were similar (75th for Group I vs. 72.5th for Group II; *p* = 0.498). However, Group I had significantly lower median percentiles for chest circumference (30 vs. 75, *p* < 0.001), body weight (50 vs. 72.5, *p* = 0.001), and height (75 vs. 90, *p* < 0.001) compared to Group II.

These results indicate statistically significant differences in growth patterns between children post-CDH surgery and their healthy peers, underscoring the need for ongoing monitoring.

In the physical performance assessment—the Wrocław test—children who underwent CDH surgeries (Group I) scored lower compared to their healthy peers (Group II) (Table 2). In Group I, the median score was Me = 194, while in Group II, it was Me = 204.5 points (*p* = 0.036) (statistically significant difference). Furthermore, a larger number of children in Group I were classified as having a low or sufficient level of physical performance, whereas participants in Group II more often achieved a good or high level of physical performance; however, this difference was not statistically significant (*p* = 0.081). Children in Group I performed significantly worse in the strength, speed, and agility tests (statistically significant differences). In the power test, the results in both groups were similar.

The scores obtained by children from Group I and Group II were referenced to the scoring scale according to the Wrocław Test. The following results were obtained.

Strength test—more children from Group I, compared to Group II, scored in this subscale with points qualifying their strength as low or insufficient, and fewer children from Group I scored in the good or very good category—both differences were statistically significant (*p* = 0.007).

Power test—no statistically significant differences between the groups (*p* = 0.846).

Speed test—no statistically significant differences between the groups (*p* = 0.136).

Agility test—significantly more participants in Group I received points in this subscale qualifying their agility as insufficient or sufficient, while noticeably fewer participants in Group I, compared to Group II, scored in the good or very good range (*p* < 0.001).

Overall score—no statistically significant differences between the groups (*p* = 0.081), although there is a trend indicating a noticeably lower physical performance of children in Group I compared to Group II (Figure 1).

The analysis of physical and motor development outcomes was conducted using age-standardized scores, enabling reliable comparisons of performance across different age groups. The results obtained in the Ozierecki Metric Scale modified by Barański confirmed the presence of significant differences in motor skills between the children in the study group and the reference group (Table 3). The chronological age of the children included in the study was, on average, 57.56 (SD = 12.66) months. Participants in Group I were statistically significantly older than those in Group II, with 61.32 (SD = 13.72) months versus 54.91 (SD = 11.27) months (*p* = 0.030). There was no statistically significant difference in the assessment of motor developmental age between the groups, with 77.00 (SD = 23.16) months in Group I versus 74.07 (SD = 20.61) months in Group II (*p* = 0.566). The acceleration of motor development did not differ statistically significantly between Group I and Group II, with 15.98 (SD = 11.59) months versus 19.16 (SD = 11.60) months (*p* = 0.204).

The linear correlation coefficient between chronological age and motor developmental age was as follows:In Group I: r = 0.93 (*p* < 0.001);In Group II: r = 0.90 (*p* < 0.001);In the overall sample: r = 0.90 (*p* < 0.001).

The differences between the groups (multivariate regression) are statistically significant, indicating that in Group I, there is a significantly smaller discrepancy between the two variables than in Group II (*p* = 0.001).

The degree of motor development acceleration among study participants—according to the Ozierecki Test modified by Barański—did not differ significantly between Group I and Group II (*p* = 0.359) (no statistically significant difference) (Figure 2).

## 4. Discussion

Parents expecting a child with congenital diaphragmatic hernia (CDH) seek answers to many questions, especially those concerning the chances of survival, prognosis, and the future development of their unborn child. The physical performance of children after CDH surgery may be inferior compared to healthy children due to factors such as the defect itself and associated complications. While searching for data on this topic in the literature, it became apparent that there are few reports describing the physical performance or motor abilities of children after CDH surgeries, and the data available are often contradictory [17,18,19,20].

According to Zaccara et al., in patients operated on for congenital diaphragmatic hernia, despite a subjective sense of good health, peak oxygen uptake (VO2max), a marker of physical performance, and overall physical performance were significantly reduced [17]. Their findings suggest that physical performance was particularly poor in children who did not engage in regular sports activities after surgery. Marven et al. found that children after CDH surgery rated their physical performance as poorer than their healthy peers and enjoyed physical activity less frequently [20]. These studies were based on spirometric measurements and exercise tests on a stationary bike. Similarly, research conducted by Koziarkiewicz et al. found that, although most patients after CDH surgery perceived their health as good, a minority reported weaker physical performance compared to peers [5]. These findings align with studies by Peetsold et al., which suggest that physical performance in this group can vary and often does not significantly differ from healthy peers [21]. Interestingly, some studies in the literature show that leading an active lifestyle, even in children after CDH surgeries, improves exercise tolerance, reduces feelings of breathlessness, and consequently increases their physical performance [20,22].

In our study, the assessment of physical performance in children from both groups was conducted using the Wrocław Test, which involved performing four tasks evaluating different aspects of physical performance. The results revealed that children in the study group scored lower in strength and agility tests, while power test scores were similar across groups. These differences resulted in slightly lower overall physical performance scores in the study group, which were statistically significant. Despite this, some children in the study group achieved high levels of physical performance, comparable to their peers. The question remains whether the poor physical performance of a large group of children after CDH surgery was due solely to the nature of the defect and the treatment they underwent, or whether environmental factors (e.g., low physical activity resulting from the overprotectiveness of caregivers) also had a significant impact on its development.

In our study, the Ozierecki Metric Scale modified by Barański was used to assess the motor skills of children. This revealed no significant differences in motor developmental age between groups, although children in the study group demonstrated slightly less motor development acceleration.

Based on these findings, it is reasonable to conclude that motor development in children after CDH surgery may be slightly weaker than in their peers.

Our study provides valuable insights regarding the perception of patients after congenital diaphragmatic hernia (CDH) surgery, both for doctors and the parents of these patients. It shows that children who have undergone surgery for this severe congenital defect do not exhibit significant differences in physical and motor abilities compared to their peers without a medical history. It is the role of doctors to make parents aware that better physical fitness will help their children succeed in school, achieve better academic results, and that having undergone CDH surgery does not mean they must forgo physical activity. This is especially important considering that participating in sports not only benefits physical health but also mental health, and moreover, improves quality of life and life satisfaction [21].

It is important to note that differences in the physical condition of children could partly be attributed to their age, which was accounted for in the statistical analysis by normalizing results relative to age-specific standards. Nevertheless, even after adjusting for age, children who underwent CDH surgery achieved slightly lower scores in some tests compared to their healthy peers. This may suggest that the impact of CDH and its treatment has a subtle influence on physical development, warranting further investigation in long-term studies.

This study has certain limitations that should be acknowledged. First, the relatively small sample size, particularly in the study group, may limit the generalizability of the findings to the broader population of children who have undergone CDH surgery. Second, the study relied on standardized physical and motor performance tests, which, although validated, may not fully capture the nuances of individual physical activity levels or environmental influences. Third, the cross-sectional design of the study provides a snapshot of the participants’ performance but does not allow for longitudinal assessment of their physical development over time. Lastly, this study did not account for potential socioeconomic or psychological factors that could influence physical performance, such as parental overprotectiveness or access to physical activity resources. Future research should consider these aspects to provide a more comprehensive understanding of the physical development of children after CDH surgery.

## 5. Conclusions

This study attempts to answer a very important question often asked by parents expecting a child with congenital diaphragmatic hernia (CDH).

Based on the results of both the Wrocław Test and the Ozierecki Metric Scale modified by Barański, it can be concluded that preschool-aged patients after CDH surgery achieve slightly lower scores in tests evaluating their physical performance and motor skills. These differences do not apply to all patients and are often minor. Despite children after CDH surgery achieving somewhat lower results in physical and motor skill tests, it seems that an important role for doctors involved in treating CDH patients is to educate the caregivers of these patients that engaging in sports is not contraindicated for this group. Furthermore, regular physical activity can significantly improve performance, leading to better physical fitness and daily functioning for these young patients.

## Figures and Tables

**Figure 1 jcm-13-07249-f001:**
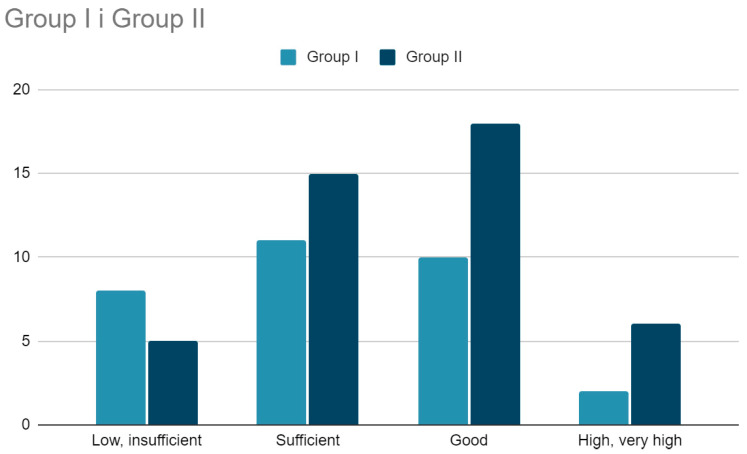
Categorization of the overall score obtained in the Wrocław test by children from Group I and Group II (*p* = 0.081).

**Figure 2 jcm-13-07249-f002:**
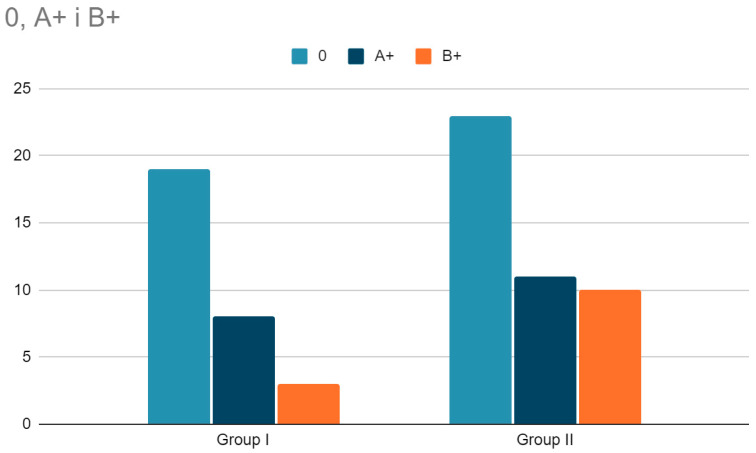
Degree of motor development acceleration according to the Ozierecki Test modified by Barański in the examined children from Group I and Group II (*p* = 0.359).

**Table 2 jcm-13-07249-t002:** Results of the Wrocław test conducted in Group I and Group II.

Analyzed Sample		Group	Statistical Parameter					Level of Statistical Significance (*p*)
			M	Me	SD	95% CI	Min.–Max.	
Strength (points)	test	Group I	39.23	38	13.49	34.28–44.17	10–72	=0.014
		Group II	46.52	49	12.59	42.69–50.35	17–72	
		Total	43.51	45	13.38	40.43–46.58	10–72	
Power (points)	test	Group I	48.68	51	19.04	41.63–55.66	0–86	=0.808
		Group II	48.02	50.50	18.17	42.50–53.55	0–81	
		Total	48.29	51	18.41	44.06–52.53	0–86	
Speed (points)	test	Group I	32.06	33	21.07	24.33–39.79	0–68	=0.027
		Group II	42.16	43.50	17.48	36.84–47.47	0–93	
		Total	37.99	40	19.56	33.49–42.49	0–93	
Agility (points)	test	Group I	57.97	51	30.84	46.65–69.28	0–124	=0.016
		Group II	62.91	60	11.98	59.27–66.55	36–108	
		Total	60.87	60	21.79	55.85–65.88	0–124	
Overall (points)	score	Group I	177.94	194	49.37	159.83–196.04	74–260	=0.036
		Group II	199.61	204.50	38.33	187.96–211.27	75–266	
		Total	190.65	195	44.24	180.47–200.83	74–266	

M (mean)—average; Me (median)—median; SD (standard deviation)—standard deviation; CI (confidence interval)—confidence interval.

**Table 3 jcm-13-07249-t003:** Comparison of motor developmental age with chronological age (in months) based on the Ozierecki Test results for children from Group I and Group II.

Age (Months)	Group	Statistical Parameter					Level of Statistical Significance (*p*)
		M	Me	SD	95% CI	Min.–Max.	
Chronological age	Group I	61.32	60	13.72	56.29–66.35	36–78	=0.030
	Group II	54.91	54.50	11.27	51.48–58.34	37–78	
	Total	57.56	56	12.66	54.65–60.47	36–78	
Motor developmental age	Group I	77.00	70	23.16	68.50–85.49	40–116	=0.566
					67.80–8		
	Group II	74.07	75.50	20.61	0.33	44–115	
	Total	75.28	75	21.59	70.31–80.25	40–116	
Age acceleration	Group I	15.68	10	11.59	11.43–19.93	0–40	=0.204
	Group II	19.16	12	11.60	15.63–22.69	6–39	
	Total	17.72	11	11.65	15.04–20.40	0–40	

M—mean; Me—median; SD—standard deviation; CI—confidence interval.

## Data Availability

The original contributions presented in the study are included in the article, further inquiries can be directed to the corresponding author/s.

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
