# Peer review of "Assessment of Physical Performance in Children After Surgery for Congenital Diaphragmatic Hernia"

_jcm, 2024, doi:10.3390/jcm13237249_

Round 1
Reviewer 1 Report
Comments and Suggestions for Authors
The authors have examined the physical performance of CDH patients after surgery when aged 3 to 6 years compared to controls. They conclude that they had reduced physical performance, in particular related to endurance and motor coordination
Comments
There is no sample size given not the reproducibility of the assessments, thus it is not possible to determine the veracity of the results
The discussion is overlong and should be reduced to any new proven results and the limitations of the study
Author Response
Comments 1: There is no sample size given not the reproducibility of the assessments, thus it is not possible to determine the veracity of the results
Response 1:
Thank you for highlighting the absence of information on sample size and assessment reproducibility. We have made the necessary corrections, highlighted in red in the "Materials and Methods" section. The changes include:
-
Addition of detailed information on sample size – the number of participants is now clearly stated for both groups: 31 children who underwent CDH surgery in the study group and 41 healthy children in the control group. This added detail provides greater clarity on the study size and its representativeness.
-
Information on reproducibility of assessments – to ensure result consistency, each test was conducted twice, and the better result was used for analysis to minimize variability. We also noted that the assessment tools used, the WrocÅ‚aw Test and the Ozierecki Metric Scale modified by BaraÅ„ski, are standard and widely used methods for evaluating physical and motor performance in children, with high reproducibility as supported by the literature.
These changes in the "Materials and Methods" section aim to enhance transparency and confirm the reliability of the results, which we hope addresses your concerns regarding the accuracy and reproducibility of the methods used.
Comment 2: The discussion is overlong and should be reduced to any new proven results and the limitations of the study
Response 2:
Thank you for your comment regarding the length of the discussion section. Following your suggestion, we have made changes highlighted in red in the "Discussion" section. To focus more on the new findings and limitations of the study, we have shortened sections not directly related to our findings.
The changes include:
-
Condensing references to previous literature – we removed some general information about prior studies that did not add essential context to our findings. Instead, we focused on discussing our results in relation to previous studies, emphasizing how these findings contribute to the existing body of knowledge.
-
Highlighting study limitations – we added a summary of the main limitations to provide a clearer perspective on interpreting the results. This addition aims to facilitate further research on physical and motor performance in children post-CDH surgery.
These changes allow for a more precise and concise discussion, aligning with your suggestion.
Please see the attachment

Reviewer 2 Report
Comments and Suggestions for Authors
Abstract requires modifications
Introduction requires more data
Methodological Biases exist
Discussion contains Results
Conclusion(s) Section must be reduced
(The Authors must see my remarks)

Author Response
Comments 1: Abstract requires modifications
Response 1:
Thank you for the suggestion to modify the abstract. We have made revisions, which are highlighted in red in the "Abstract" section, to improve clarity and accuracy.
The changes include:
-
Adding subsections to the abstract – we divided the abstract into clearly defined sections: Objective, Aim, Methods, Results, and Conclusions. This structure allows for a clearer presentation of the study’s purpose, methods, results, and conclusions, in line with scientific publication standards.
These changes aim to provide a more concise and transparent presentation of the study’s content in accordance with your suggestions.
Comments2: Introduction requires more data
Response2:
Thank you for your feedback on the Introduction. We have revised this section to include additional background information, which is highlighted in red for easy reference.
The changes made include:
-
Expanded description of long-term complications – we added details on chronic respiratory issues, neurodevelopmental impairments, and growth delays, which can persist into adolescence, to provide a more comprehensive understanding of the ongoing challenges faced by children with CDH.
-
Further discussion on physical and motor development – we included findings from recent studies indicating that children with CDH may experience delays in physical milestones, lower endurance, and reduced motor coordination compared to their healthy peers, which helps contextualize the importance of our study.
-
Rationale for long-term assessment – we emphasized the need for further research on physical and motor development in children post-CDH surgery, particularly with respect to quality of life and potential benefits of targeted rehabilitation.
These changes aim to provide additional context and justification for the study, addressing the points raised in your comment.
Comment3: Methodological Biases exist
Response3:
Thank you for highlighting the absence of information on methological biases. We have made the necessary corrections, highlighted in red in the "Materials and Methods" section. The changes include:
-
Addition of detailed information on sample size – the number of participants is now clearly stated for both groups: 31 children who underwent CDH surgery in the study group and 41 healthy children in the control group. This added detail provides greater clarity on the study size and its representativeness.
-
Information on reproducibility of assessments – to ensure result consistency, each test was conducted twice, and the better result was used for analysis to minimize variability. We also noted that the assessment tools used, the WrocÅ‚aw Test and the Ozierecki Metric Scale modified by BaraÅ„ski, are standard and widely used methods for evaluating physical and motor performance in children, with high reproducibility as supported by the literature.
These changes in the "Materials and Methods" section aim to enhance transparency and confirm the reliability of the results, which we hope addresses your concerns regarding the accuracy and reproducibility of the methods used.
Comments4: Discussion contains Results
Response4:
Thank you for noting that results were included in the Discussion section. We have removed these results from the Discussion, and the deleted text is highlighted in dark blue for easy reference.
These changes ensure that the Discussion section focuses solely on the interpretation of findings and aligns with standard structure requirements.
Comments5: Conclusion(s) Section must be reduced
Response 5:
Thank you for your suggestion to reduce the Conclusions section. We have made revisions to streamline this section, focusing on the main findings and recommendations for supporting physical activity in patients post-CDH surgery. The reduced text is highlighted in dark blue to facilitate review.
These changes aim to create a more concise conclusion that directly addresses the study’s primary outcomes and recommendations.
Comments 5: Remarks
Response to Reviewer’s Comment:
We have made revisions in the Methodology section to enhance clarity and detail. These changes are highlighted in green for easy reference.
We hope these modifications address your comments and improve the overall transparency of our methodology.
Thank you for your question regarding the statistical methodology. We acknowledge the importance of selecting appropriate models and appreciate your consideration of logistic regression.
In this study, we chose to analyze categorical variables using Chi-square independence tests, Fisher’s exact test (for small cell counts), and log-linear analysis. For measurable variables, we applied multifactorial ANOVA for normally distributed data and generalized linear models with robust standard errors for variables deviating from normality. These multivariate models were controlled for key covariates such as gender, age, body weight, and height, which we believe effectively address potential confounders.
Given the structure of our data and the study’s objectives, these statistical methods provide a robust framework for assessing group differences and associations. Therefore, we determined that these analyses were sufficient without necessitating logistic regression.
Thank you for your question regarding the model of statistical significance. We have clarified that a p-value threshold of < 0.05 was used as the criterion for statistical significance across all analyses, including multifactorial ANOVA, generalized linear models, Chi-square tests, and Fisher’s exact tests. This threshold is commonly used in scientific research to indicate statistical significance, allowing comparability with other studies.
This information is specified in the Statistical Analysis section of the manuscript.
We have revised this section to focus more directly on the study's main findings and limitations. The text marked for removal is highlighted in dark blue in the Discussion section for easy reference.
These changes aim to create a more concise discussion that aligns with your feedback.
The subject of this study is challenging to discuss due to its complexity, which has limited the number of available references. Additionally, this topic has not been widely addressed in the literature, which further restricts the scope for citations.
We hope this explanation clarifies the limited number of references in the manuscript.
We have revised the References section in accordance with your recommendations.
Please see the attachment

Reviewer 3 Report
Comments and Suggestions for Authors
The aim of this study was to assess the physical performance, endurance, and motor coordination of children after congenital diaphragmatic hernia (CDH) surgery and compare their results with those of a group of healthy peers. The study aims to provide data that may contribute to the development of recommendations for long-term care of patients after CDH surgery.
The manuscript is well written, but some revisions are needed to make the research more relevant and the article more readable.
Anthropometric parameters and assessment of physical development of the study groups should be presented. The conditions (indoor/outdoor) of the study should be indicated.
The safety of using a 1 kg ball strength test in children aged 3-6 years should be clarified. Given the surgeries performed, the use of heavy weights may be contraindicated in such patients, and for children this may be too much of a strain.
The need for Table 1 should be explained because the study included children aged 3-6 years. Table 1 does not provide information on the assessment of children aged 3 years. Information about children over 7 years of age does not correspond to the purpose of the study.
When presenting the results, there is no need to average the ages of the participants, since children of different ages differ in their physical condition and the development of motor and neuropsychic skills.
To correctly evaluate the results, standardization of tests and normalization are required in accordance with the age of the study participants.
Pay attention to the use of abbreviations.
The abstract should be edited to better suit the research conducted.
The list of references contains many references to studies conducted more than 10 years ago, which reduces the relevance and scientific value of the results. The list of citations needs to be revised.
Author Response
Comments 1: Anthropometric parameters and assessment of physical development of the study groups should be presented. The conditions (indoor/outdoor) of the study should be indicated.
Response 1:
Response to Reviewer’s Comment:
Thank you for your suggestion to clarify methodological details and present anthropometric characteristics. We have made the following revisions:
-
Methodology – We expanded the section to include the type and method of anthropometric measurements, detailing parameters such as head and chest circumference, body weight, and height. Additionally, we specified that all physical performance assessments were conducted indoors under controlled conditions to ensure consistency and minimize environmental influences on test results. These changes are highlighted in red in the "Materials and Methods" section.
-
Results – Additional details on the demographic and anthropometric characteristics of the studied children have been included, providing a clearer comparison between Group I and Group II and highlighting differences in growth patterns. This updated information is highlighted in red in the "Results" section under "Demographic and Anthropometric Characteristics of the Studied Children."
We hope these adjustments meet your expectations and clarify the study design and findings.
Comments2: The safety of using a 1 kg ball strength test in children aged 3-6 years should be clarified. Given the surgeries performed, the use of heavy weights may be contraindicated in such patients, and for children this may be too much of a strain.
Response2:
We would like to clarify that this test is a standard assessment for evaluating upper body strength in children within this age range and has been widely validated in pediatric settings.
In our study, all assessments were conducted under the supervision of trained professionals who closely monitored each child’s comfort and tolerance throughout the test. Children displaying any signs of discomfort or strain were instructed to stop immediately to ensure their well-being. Furthermore, children were screened to confirm that they met the criteria for safely participating in this type of physical assessment.
These clarifications have been added to the Materials and Methods section (Strength test subsection), highlighted in red for ease of reference. We hope this addresses your concerns about the safety of the strength test used in our study.
Comments3: The need for Table 1 should be explained because the study included children aged 3-6 years. Table 1 does not provide information on the assessment of children aged 3 years. Information about children over 7 years of age does not correspond to the purpose of the study.
Response3:
Thank you for noting the need for clarification regarding the participation of three-year-old children. We agree that this should be addressed, and we have added a note in the text to clarify their involvement. Specifically, we have explained that three-year-olds actively participated in the study using exercises designated for four-year-olds, but their results were not included in the final statistics due to potential reliability concerns.
This addition is highlighted in red in the "Materials and Methods" section, under the paragraph titled "Ozierecki Metric Scale, modified by Barański."
Thank you again for your valuable observation.
Comments4: When presenting the results, there is no need to average the ages of the participants, since children of different ages differ in their physical condition and the development of motor and neuropsychic skills.
Response4: We have removed the average age data from the results section to better reflect the developmental differences across age groups. The text marked for removal is highlighted in dark blue in the article.
Comments5: To correctly evaluate the results, standardization of tests and normalization are required in accordance with the age of the study participants.
Response5:
We have clarified in the Materials and Methods section that all test scores were normalized according to age-specific standards, and we used recognized measurement tools that account for developmental differences in children. Additionally, we have highlighted in the Results section that all performance outcomes were analyzed with age-standardized scores, ensuring reliable comparisons across age groups.
These changes are highlighted in green in the "Materials and Methods" and "Results" sections for your review. We hope these additions address your concern and enhance the clarity of our approach.
Comments6: Pay attention to the use of abbreviations.
Response6: We have reviewed the manuscript to ensure that all abbreviations are defined at their first mention and used consistently throughout. Additionally, unnecessary abbreviations have been removed to improve readability.
Comments7: The abstract should be edited to better suit the research conducted.
Response7:
We have revised it to align more closely with the research conducted, following the standard structure for scientific abstracts. Specific sections, including Objective, Methods, Results, and Conclusions, have been clarified to better reflect the study’s purpose, key methodologies, main findings, and implications.
These changes are highlighted in red in the Abstract section for easy reference.
Comments8: The list of references contains many references to studies conducted more than 10 years ago, which reduces the relevance and scientific value of the results. The list of citations needs to be revised.
Response8:
The subject of this study is challenging to discuss due to its complexity, which has limited the number of available references. Additionally, this topic has not been widely addressed in the literature, which further restricts the scope for citations.
We hope this explanation clarifies the limited number of references in the manuscript.
We have revised the References section in accordance with your recommendations.
Please see the attachment

Round 2
Reviewer 1 Report
Comments and Suggestions for Authors The authors have modified to meet my comments, except in the discussion. Indeed they have added to the discussion rather reduce it and I do not see information on the limitations of their study
Author Response
Comments 1: The authors have modified to meet my comments, except in the discussion. Indeed they have added to the discussion rather reduce it and I do not see information on the limitations of their study
Response 1:
In response to your feedback, we have implemented the following changes:
-
Shortened Discussion Section:
- Sections of the discussion that were overly detailed or repetitive were removed. The deleted text has been marked with a navy-blue highlight for your reference.
- Key findings have been preserved, and the revised, condensed text has been underlined to clearly indicate the updated version. These changes ensure that the discussion is more concise and focused on the study's essential aspects.
-
Addition of Study Limitations:
- We have added a paragraph addressing the study's limitations at the end of the Discussion section. This paragraph highlights key constraints, such as the relatively small sample size, the cross-sectional study design, and the potential influence of unmeasured factors like environmental or psychological influences.
We hope these changes address your concerns and improve the clarity and quality of the manuscript. Thank you once again for your constructive feedback. Please see the attachment.

Reviewer 3 Report
Comments and Suggestions for Authors
The aim of this study was to assess the physical performance, endurance, and motor coordination of children after congenital diaphragmatic hernia surgery and compare their results with those of a group of healthy peers. The study aimed to provide data that may contribute to the development of recommendations for long-term care of patients after congenital diaphragmatic hernia (CDH) surgery.
The manuscript has been improved, it is well written, but some revisions are needed to make the research more relevant and the article more readable. The data should be presented more correctly. Average anthropometric and age data are still presented, however, children of different ages differ in their physical condition. Therefore, it is incorrect to average these parameters.
The discussion contains many repetitions of the results obtained. Instead, critical reflection on the findings is required, as well as a presentation of the limitations of the study.
The list of references contains many references to studies conducted more than 10 years ago, which reduces the relevance and scientific value of the results. PubMed has many studies that can be referenced. The list of citations needs to be revised.
The article needs to be formatted according to the journal's rules. There are typos in the text that need to be corrected.
Author Response
Comments 1: The data should be presented more correctly. Average anthropometric and age data are still presented, however, children of different ages differ in their physical condition. Therefore, it is incorrect to average these parameters.
Response 1:
Thank you for your thoughtful comments regarding the averaging of anthropometric and age data and its potential impact on the interpretation of our results. We understand your concern that averaging these parameters might not fully reflect the variability in physical condition among children of different ages. After careful consideration, we have chosen not to perform further stratification of the data for the following reasons:
-
Normalization and Age-Specific Analysis:
- The physical performance and anthropometric data were already normalized according to age-specific standards, ensuring that comparisons were made within an appropriate developmental context.
- This approach allows for a fair assessment of differences between the study and control groups without introducing excessive fragmentation of the dataset, which could reduce statistical power given the sample size.
-
Impact of Age Considered in Statistical Analysis:
- In the Statistical Analysis section, we clarified that the variability related to age was controlled by analyzing data in the context of age-specific norms and employing statistical models that account for age differences.
- This ensures that any observed differences are not confounded by age-related variability, thus addressing the core of your concern.
-
Added Explanation in the Discussion Section:
- To further address your feedback, we included a new paragraph in the Discussion section to emphasize the role of age differences in physical performance and how they were accounted for in our analysis. We also highlighted that, even after adjusting for age, children in the study group tended to achieve slightly lower scores in some tests compared to their healthy peers, suggesting subtle effects related to CDH and its treatment.
These revisions have been clearly marked with underlining in the Statistical Analysis and Discussion sections for your convenience. We believe that this approach maintains the integrity of the data while addressing your concerns regarding the presentation and interpretation of the results.
We hope these changes meet your expectations and further clarify the methodology and findings of our study. Thank you once again for your valuable insights, which have strengthened the manuscript.
Comments 2: The discussion contains many repetitions of the results obtained. Instead, critical reflection on the findings is required, as well as a presentation of the limitations of the study.
Response 2:
Thank you for your constructive feedback regarding the discussion section. In response to your comments, we have made the following changes:
-
Text Simplification and Improved Clarity:
- The discussion has been carefully revised to reduce unnecessary repetitions of the results. Instead, we focused on providing critical reflections and interpretations of the findings in relation to existing literature.
- These revisions make the text more concise and improve its overall clarity and readability.
-
Incorporation of Study Limitations:
- A new paragraph has been added to the end of the Discussion section to explicitly address the limitations of the study. This includes considerations such as the relatively small sample size, the cross-sectional design, and potential environmental or psychological factors that were not accounted for.
-
Visual Marking of Changes:
- The removed text is highlighted with a navy-blue marker, while the newly added content is underlined for your convenience.
We are confident that these changes address your concerns and improve the quality of the manuscript by ensuring a more focused and reflective discussion.
Comment 3: The list of references contains many references to studies conducted more than 10 years ago, which reduces the relevance and scientific value of the results. PubMed has many studies that can be referenced. The list of citations needs to be revised.
Comments 3: The list of references contains many references to studies conducted more than 10 years ago, which reduces the relevance and scientific value of the results. PubMed has many studies that can be referenced. The list of citations needs to be revised.
Response 3:
We understand the importance of citing up-to-date and relevant literature to enhance the scientific value of our study. In response to your suggestion, we have reviewed the reference list and updated it wherever possible to include more recent studies, particularly those available on PubMed.
The updated references are highlighted with underlining in the References section for your convenience. These changes ensure that the manuscript reflects the most current scientific knowledge while maintaining key citations that provide foundational context for the study.
Comments 4: The article needs to be formatted according to the journal's rules. There are typos in the text that need to be corrected.
Response 4: In response to your comment, we have carefully proofread the text and corrected all identified typos and inconsistencies.
Please see the attachment
